# High-Pressure Inactivation of *Bacillus cereus* in Human Breast Milk

**DOI:** 10.3390/foods12234245

**Published:** 2023-11-24

**Authors:** Miroslava Jandová, Michaela Fišerová, Pavla Paterová, Lucie Cacková, Pavel Měřička, Jan Malý, Marian Kacerovský, Eliška Kovaříková, Jan Strohalm, Kateřina Demnerová, Jana Kadavá, Hana Sýkorová, Radomír Hyšpler, Dana Čížková, Aleš Bezrouk, Milan Houška

**Affiliations:** 1Tissue Bank, University Hospital Hradec Králové, 500 05 Hradec Králové, Czech Republic; michaela.fiserova@fnhk.cz (M.F.); pavel.mericka@fnhk.cz (P.M.); 2Department of Histology and Embryology, Faculty of Medicine in Hradec Králové, Charles University, 500 03 Hradec Králové, Czech Republic; cizkovad@lfhk.cuni.cz; 3Department of Clinical Microbiology, University Hospital and Faculty of Medicine in Hradec Králové, Charles University, 500 05 Hradec Králové, Czech Republic; pavla.paterova@fnhk.cz (P.P.); lucie.cackova@fnhk.cz (L.C.); 4Department of Pediatrics, University Hospital Hradec Králové, 500 05 Hradec Králové, Czech Republic; jan.maly@fnhk.cz; 5Biomedical Research Center, University Hospital Hradec Králové, 500 05 Hradec Králové, Czech Republic; marian.kacerovsky@fnhk.cz; 6Food Research Institute Prague, 102 00 Prague, Czech Republic; eliska.kovarikova@vupp.cz (E.K.); jan.strohalm@vupp.cz (J.S.); milan.houska@vupp.cz (M.H.); 7Department of Biochemistry and Microbiology, University of Chemistry and Technology Prague, 166 28 Prague, Czech Republic; katerina.demnerova@vscht.cz (K.D.); jana.kadava@vscht.cz (J.K.); hana.sykorova@vscht.cz (H.S.); 8Department of Clinical Biochemistry and Diagnostics, University Hospital Hradec Králové, 500 05 Hradec Králové, Czech Republic; radomir.hyspler@fnhk.cz; 9Department of Medical Biophysics, Faculty of Medicine in Hradec Králové, Charles University, 500 03 Hradec Králové, Czech Republic; bezrouka@lfhk.cuni.cz

**Keywords:** pressurization, *Bacillus cereus*, human breast milk, inoculation

## Abstract

Although Holder pasteurization is the recommended method for processing breast milk, it does affect some of its nutritional and biological properties and is ineffective at inactivating spores. The aim of this study was to find and validate an alternative methodology for processing breast milk to increase its availability for newborn babies and reduce the financial loss associated with discarding milk that has become microbiologically positive. We prepared two series of breast milk samples inoculated with the *Bacillus cereus* (*B. cereus*) strain to verify the effectiveness of two high-pressure treatments: (1) 350 MPa/5 min/38 °C in four cycles and (2) cumulative pressure of 350 MPa/20 min/38 °C. We found that the use of pressure in cycles was statistically more effective than cumulative pressure. It reduced the number of spores by three to four orders of magnitude. We verified that the method was reproducible. The routine use of this method could lead to an increased availability of milk for newborn babies, and at the same time, reduce the amount of wasted milk. In addition, high-pressure treatment preserves the nutritional quality of milk.

## 1. Introduction

Human breast milk (HBM) is valuable. It is obtained from female donors and is used to feed children in the first days of life who, for various reasons, cannot be fed by their own mother. HBM is subject to strict microbiological monitoring and any unremovable contamination is grounds for the disposal of the milk. Contamination with bacterial spores is a serious problem because the spores cannot be removed using conventional methods [1,2,3,4].

The practice of our Human Milk Bank has shown that the most common spore-forming microbe present in breast milk is *B. cereus*, which accounted for up to 64% of all positive microbial findings. There is also the risk of *B. cereus* spores germination during the warming of pasteurized milk [5,6].

Holder pasteurization (62.5 °C/30 min), which is used to treat breast milk in most Human Milk Banks, cannot effectively eliminate all germs, and after pasteurization, 7–14% of milk is still culture-positive. Contaminated milk must then be discarded from further use according to the standard [1]. Additionally, the Holder method also affects the biological quality of human milk. A number of studies dealing with the analysis of immunological components have confirmed that HPT is more suitable for its conservation than conventional Holder pasteurization [1,7,8,9,10].

Currently, other methods for the treatment of HBM that reduce microorganisms, such as UV-treatment [11,12,13,14], high-temperature treatment [15,16,17,18,19,20,21], thermo-ultrasonication [12,22,23], high-intensity pulsed electric field [24], and high-pressure treatment, are being studied [1,2,12]. However, these methods have not yet been introduced into routine practice in Human Milk Banks [1,2].

High-pressure treatment (HPT) represents an option that should preserve the high quality of milk while also removing resistant spores. The high-pressure inactivation of spores is commonly used to eliminate spores from food products or other materials [1,2,25,26,27,28,29,30,31]. HPT is an effective method for inactivating *B. cereus* since it penetrates the cell wall and membrane, causing irreversible damage to the cell structure and function. However, it is important to optimize pressure and time parameters to ensure the complete inactivation of microorganisms while minimizing damage to milk quality [32]. Typically, the process involves subjecting spores to high pressure, typically between 100 and 800 MPa, for specific periods of time; this process is effective against a wide range of spores, including those of *B. cereus* and *Clostridioides difficile* [33].

The advantage of high-pressure inactivation is that it does not require the use of chemicals or heat that could impair the quality of the product (including HBM) and thus, can be used to treat products sensitive to heat or chemicals. On the other hand, high-pressure inactivation may not be suitable for all types of products, since some can be damaged by high pressures. It is important to carefully evaluate the product and the specific application before its use [34]. The current goal of HPT is to optimize these methods in order to achieve the best possible results in terms of nutrient retention and microbiological safety [9].

Demazeau et al. (2018), after applying various optimization tests, defined the conditions under which all vegetative forms and bacterial spores (such as *B. cereus*) were inactivated. The optimal parameters were the following: pressure 350 MPa, temperature 38 °C, treatment rate = 1 MPa·s^−1^, for four cycles with cycle duration 5 min each, and latency time (with normal pressure) between each cycle—5 min. Additionally, they found that the bioactivity of many main components, including lipase, α-lactalbumin, casein, lysozyme, lactoferrin, and immunoglobulin IgAs, was preserved [2]. Fekraoui et al. investigated the advantages of using cycled HPT compared to continuous HPT on the inactivation of *Bacillus subtilis* (*B. subtilis*) and *B. cereus* spores [35]. In his review work, Billeaud (2021) proposed to use high hydrostatic pressure (HHP) with four pressure cycles in the range of 50–150 MPa to promote the germination of *B. cereus* followed by a pressure of 350 MPa to kill 10^6^ CFU/mL of *B. cereus* spores while retaining 80–100% of lipase, lysozyme, and lactoferrin activity, and 64% of immunoglobulin IgA [1]. Furukawa et al. (2021) measured the germinating and inactivating effects of cycled HPT (using six cycles of 5 min compression followed by rapid decompression) compared with continuous pressure using heat sensitivity (i.e., 70 °C/30 min). The results showed that compression could initiate spore germination, and rapid decompression could inactivate germinated spores [25].

Hayakawa et al. measured the effect of high pressure on thermoduric (resistant to high-temperature heat treatment) spores of *Bacillus stearothermophilus,* comparing two modes: Mode I = 800 MPa/60 °C/60 min and Mode II 800 = MPa/room temperature/60 min. Mode I resulted in a decrease in the spore count (from 10^6^ to 10^2^ CFU per mL). Cycled pressurization (i.e., six cycles, 5 min each) of 400 MPa/70 °C produced similar results to Mode 1, i.e., spore counts decreased from 10^6^ to 10^2^, while cycles using 600 MPa produced complete sterilization [26].

Obaidat et al. reported the effect of using moderate hydrostatic pressure (40–140 MPa) at a moderate temperature (37–58 °C) to inactivate the spores of *B. subtilis*. The results showed that spore inactivation was exponentially proportional to the time exposed to pressure; pressures below 100 MPa and temperatures of 60 °C led to spore inactivation [27]. Doona et al. devised a “quasi-chemical” model for bacterial spore germination dynamics using HPT, which helped to promote effective reductions in bacterial spores [33].

The purpose of our research was to compare two methods of high-pressure inactivation-pressurization in cycles vs. continuous pressurization-performed on one device, using identical samples and reducing the number of spore-forming microbes, thus reducing the amount of discarded breast milk associated with it. The novelty of our work consists in verifying both procedures on a large set of data. To our best knowledge, there is not study on the same material with the same microorganisms using a four-peak high-pressure technology and a technology that used only one peak with the same pressure holding time as the individual peaks combined.

## 2. Materials and Methods

In our work, we used the findings of Demazeau et al. and Fekraoui et al. and verified the effectiveness of these methods on a larger data set [2,35]. In the case of pressurization in cycles, we also analyzed the effectiveness of the method for inactivating spores after each completed cycle. For testing in eight separate experiments, isolates of *B. cereus* came directly from breast milk (28 samples from the University Hospital Hradec Králové—UH HK), and also from the Laboratory of the University of Chemistry and Technology Prague (80 samples, a collection strain of *B. cereus* CCM 869 (WDCM 0001)), Faculty of Food and Biochemical Technology (UCT). Control blanks were prepared for each experiment. Figure 1 shows the algorithm of the entire set of experiments. A total of eight experiments were conducted, whereas experiments No. I–V were performed with milk inoculated at UCT and experiments No. VI–VIII were carried out with milk inoculated at UH HK.

### 2.1. Sample Preparation—Inoculation with B. Cereus Spores at UCT

#### 2.1.1. Preparation of the *B. cereus* Spore Suspension [36]

For the inoculation of the breast milk samples, the reference strain *B. cereus* CCM 869 (WDCM 0001) was used. The preparation of the spore suspension took place in two phases. First, the strain of *B. cereus* from the original gelatin disk was grown in Brain Heart Infusion broth (BHI; Merck, Darmstadt, Germany) for 48 h at 30 °C. After initial multiplication, cells were centrifuged 6000 g/10 min (Rotanta 460R, Andreas Hettich GmbH & Co. KG, Tuttlingen, Germany) and the pellets were resuspended in a sterile physiological solution. This suspension was then incubated at 30 °C to induce sporulation. The total number of *B. cereus* cells and the concentration of spores were continuously checked.

#### 2.1.2. *B. cereus* Spores Concentration Determination [37]

To determine the concentration of *B. cereus*, a 1 mL aliquot was taken from the suspension. The total number of *B. cereus* was determined by plating on Tryptone Soya Agar (TSA, Oxoid, Hants, UK). To determine the number of spores, cells in the vegetative state were eliminated by a heat treatment of 75 °C for 11 min. This was followed by decimal dilution and plating on TSA medium. After 5 days, the required concentration of 10^5^–10^8^ spores/mL was reached (specific numbers are presented in the tables for individual experiments).

#### 2.1.3. Preparation of Samples for High-Pressure Inactivation at UCT [37]

Breast milk samples were then inoculated with a suspension of spores of known concentration. First, 1 mL of sample suspension was added to 100 mL of thawed breast milk. After inoculation, the milk samples were divided into 2 bags of 50 mL each (NUK, Dolní Bousov, Czech Republic). A control milk sample (BLANK) was also inoculated in the same way. To verify the initial spore counts in the samples, a 1-mL aliquot was taken from several randomly selected bags, and after heat treatment (75 °C/11 min), was transferred to selective Mannitol Egg Yolk Polymyxin agar (hereafter MYP, Merck, Darmstadt, Germany). The use of selective agar eliminates any accompanying microflora in the milk and allows for the clear differentiation of typical *B. cereus* colonies based on the unique appearance of the colonies on the agar surface (dull pink colonies with a zone of precipitation). Thus, the initial concentration of spores in a 1 mL milk sample was experimentally verified. The number always corresponded to a 100-fold dilution of the suspension used for inoculation (100 mL of milk + 1 mL of initial suspension).

### 2.2. Sample Preparation—Inoculation with B. cereus Spores at UH HK

#### 2.2.1. Preparation of *B. cereus* Suspension from Collected Clinical Isolates [38]

For this experiment, we used a *B. cereus* strain derived from common clinical samples isolated in our previous research [5,6]. These strains were revitalized and cultivated on Columbia agar (Oxoid, Ltd., Hampshire, UK) for 24 h to achieve typical quantities of *B. cereus*. Using a densitometer, suspensions with a base of 0.5 McF (McFarland) were prepared, followed by a 10-fold dilution.

#### 2.2.2. Preparation of Samples for High-Pressure Inactivation at UH HK [38]

Frozen 100 mL bottles of HBM were thawed in the refrigerator for 24 h, then stabilized in a 22 °C water bath. Before inoculation, the initial concentration of *B. cereus* was determined in all bottles using the cultivation on blood agar (18–24 h at 35 ± 2 °C). The milk samples (in 100 mL bottles) were inoculated with the *B. cereus* suspension prepared as described above, and each sample was divided into two parallel aliquots and placed into bags (NuK, Dolní Bousov, Czech Republic). Samples of 0.5 mL were taken from each inoculated bottle and cultivated on blood agar for 18–24 h at 35 ± 2 °C to determine the exact concentration of bacteria after inoculation. After a 24-h culture, i.e., the following day, we counted the resulting quantity of *B. cereus* on the agars before and after inoculation.

### 2.3. High-Pressure Treatment Procedure

The samples with a volume of 100 mL were treated in plastic sealed bags (NUK, Dolní Bousov, Czech Republic) (Figure 2) using a high-pressure isostatic press CYX 6/103 (Žďas join-stock company, Žďár nad Sázavou, Czech Republic) (Figure 3), with a chamber volume of 2 L, tempered to 38 °C.

#### 2.3.1. Pressurization in Cycles

The samples were subjected to increasing pressure for approximately 5 min during each cycle. After reaching a pressure of 350 MPa, there was a 5-min hold, followed by rapid depressurization. The entire cycle, each lasting 10 min, was repeated four times. The sample was then cooled to 5–8 °C and sent for analysis. Figure 4 shows, in detail, the pressure changes during one cycle, without a pause after decompression. Figure 5 illustrates, in detail, the pressure increase during the first 5 min of the cycle, i.e., pressure was increased in 3-s increments, followed by a 27-s period of equalization. An equalization period was included to achieve an overall average pressurization rate of 1 MPa per second. (Note that Figure 4 only shows the pressure changes during the first 2 min of a single cycle).

After all four cycles were completed, the samples were subjected to microbiological analysis (Experiments No. I–III). The following experiments (No. IV and No. V) were performed to show how the microbial load decreased after each completed cycle. Individual samples were subjected to a pressure of 350 MPa according to the schedule presented in Table 1. The results of this experiment were also compared with continuous pressurization (i.e., without cycles) at 350 MPa/38 °C/20 min (two samples, No. 9 and 10). Sample BLANK 1 and BLANK 2 underwent the same temperature history but were not pressurized.

#### 2.3.2. Continuous Pressurization [35]

Samples were subjected to increasing pressure over 5–6 min. After reaching a final pressure of 350 MPa/38 °C, the samples were held at these conditions for 20 min; after depressurization, the samples were cooled to 5–8 °C and sent for analysis.

### 2.4. Microbial Analysis of Samples

#### 2.4.1. Microbial Analysis Performed at UCT [36]

After the HPT of the milk samples, the total number of *B. cereus* in all the samples was determined. A standard methodology was used: a ten-fold dilution of the sample, spreading 200 µL on the surface of selective MYP agar (elimination of accompanying microflora), and the calculation of CFU/mL.

The control samples of milk, i.e., without pressure treatment (BLANK), were processed in the same way. In addition, the number of spores present in the control samples was determined (thermal heating and subsequent spreading on MYP, see previously described procedure). The goal was to verify the effect of time and temperature during sample handling on both the total number of *B. cereus* and the number of spores that can germinate into vegetative forms during the process. The determination of the effectiveness of the pressure treatment could be distorted if there was a significant reduction in the number of spores during the sample handling of the samples.

#### 2.4.2. Microbial Analysis Performed at UH HK [38,39]

It was assumed that only spores survive pressure treatment, so the resulting number corresponds to the number of spores. The number of spores was verified using a quantitative method; the inoculation of the milk samples on blood agar (incubation for 18–24 h/at 35 ± 2 °C) [39] and also by inoculation into thioglycolate broth followed by inoculation on blood agar (incubation for 18–24 h/at 35 ± 2 °C), with results of either “positive” or “negative” [38]. BLANKs underwent the same thermal history but without pressurization.

### 2.5. Statistic Methods

The data were statistically evaluated using MS Excel 2016 (Microsoft Corp., Redmond, WA, USA) and NCSS 10 statistical software (2015, NCSS, LLC., Kaysville, UT, USA, and available online: ncss.com/software/ncss (accessed on 21 April 2023)). The data from normally distributed populations with more than 10 results in a test group were described using the mean and standard deviation of the sample ($\bar{x}$ ± SD), while the other data were described using the median and the first and third quartiles of $\tilde{x}$ (1st Q, 3rd Q). For the effectiveness of the individual pressurization methods used for the data from the samples inoculated at UCT, the Equal-Variance *T*-Test at α = 0.05 was used, and for the data from both pressure methods for the samples inoculated at UH HK, the Wilcoxon Signed-Rank Test at α = 0.05 was used.

We also compared residual microbial contamination for both pressurization methods using the Equal-Variance *T*-Test; to adjust for multiple comparisons and keep the α at level 0.05, the Bonferroni correction was used. Thus, the resulting α for a single comparison was 0.017.

## 3. Results

### 3.1. Efficiency Comparison of Both Pressure Methods for Samples Inoculated at UCT

Table 2 contains basic descriptive statistics for the data obtained from both methods of pressurization in two experiments (No. I and No. II). We statistically proved that pressurizing in cycles led to a significantly (α = 0.05; *p* < 0.001) lower number of spores than continuous pressurization, i.e., pressurizing in cycles appeared to be more effective at inactivating *B. cereus* spores. The results of Experiment I and II are also presented in Figure 6 and Figure 7.

We also selected six representative samples from the entire data set for which a complete set of measurements was performed, and we determined the percentage decrease in *B. cereus* (CFU/mL) compared to the nominal value (Table 3). We demonstrated that both methods led to statistically significant decreases in microbial contaminants (α = 0.017, *p* = 0.365). The slightly higher efficiency of pressurization cycles, i.e., method P1, is shown in Figure 8, which, however, was not found to be statistically significant.

### 3.2. Efficiency Comparison of Both Pressure Methods for Samples Inoculated at UH HK

The breast milk samples were inoculated with a suspension of the *B. cereus* clinical strain isolate in an initial quantity of an average of 92 ± 64 CFU/mL, median 72 (60; 97) CFU/mL. Table 4 summarizes the basic descriptive statistics for data after both methods of pressurization. Because the median and quartile values were zero, we also present the mean values for informative purposes. The method of pressurizing in cycles has been shown to be significantly (α = 0.05, *p* = 0.024) more effective at inactivating *B. cereus* spores.

### 3.3. Results of Microbial Analysis of Samples with Successive Sampling: Individual Cycles vs. Continuous Pressurization

Table 5 shows the number of *B. cereus* spores after one, two, three, and four pressurization cycles and after continuous pressurization, including values for BLANK 1 and BLANK 2 (i.e., temperature but no pressure). The initial inoculated *B. cereus* spore count in milk was of the order of 10^4^ CFU/mL (Experiment IV) and 10^6^ CFU/mL (Experiment V). The presented results showed that after the application of the first and second cycle, the number of spores decreased to 10^3^ CFU/mL, and after the third and fourth cycle, to 10^2^ CFU/mL. The pressurization applied in cycles thus inactivated spores most effectively after the third cycle. Continuous pressure application was able to only decrease counts to 10^3^ CFU/mL.

### 3.4. Pressurization Methods over Time

Figure 9 and Figure 10 show the time courses of pressurization in summary for all runs. Figure 9 is a graph of the pressurization cycles and Figure 10 shows continuous pressurization. The figures show that the time courses of the individual pressures were comparable in repeated experiments. Individual graphs of each experiment are presented separately in the Appendix A. The beginning of each cycle cannot be the same due to the design of the device, which depends on where the multiplier piston stopped during the previous pressurization cycle.

## 4. Discussion

It is well known that HPT activates the receptors of the inner membrane of spores, which leads to their germination. Subsequently, the high pressure causes the destruction of some vital enzymes and disturbs genetic mechanisms, such as transcription and translation, leading to the inactivation of spores [9,29,40]. The higher efficiency of pressurization in cycles compared to continuous ones aligns with the phenomenon known as sensitization that can occur with certain stress treatments, including pressure cycles in the context of bacterial spores. Sensitization refers to the increased susceptibility of microbial cells to subsequent stress or damage after exposure to an initial sublethal stress. In the case of pressurization in cycles compared to continuous treatment, the hypothesis is that the first pressure cycle may trigger the initiation of spore germination or activate cellular processes that make the spores more responsive to environmental influences. As a result of the initial pressure treatment, the spores may become more sensitive to external stresses, making them more susceptible to subsequent pressure cycles. Subsequent pressure cycles can then have a more significant impact on the already sensitized spores. This cumulative effect enhances the overall efficacy of the pressurization process [40,41,42]. The presented data show the effectiveness of high-pressure treatment on *B. cereus* spores, i.e., reducing counts to less than 1% of the initial values (Table 3, Figure 8). The use of pressurization in cycles led to lower CFU numbers than continuous pressurization, i.e., pressurization in cycles appeared to be more effective for the inactivation of *B. cereus* spores. Overall, we observed that pressurization led to a reduction in the number of *B. cereus* spores by three to four orders of magnitude, which was similarly observed by other authors, including Hayakawa et al. [26]. Demazeau et al. reported a reduction of up to six orders for both *Staphylococcus aureus* and *B. cereus* [2].

Although the input samples inoculated with *B. cereus* collection strain at UH HK had the quantities commonly found in clinical isolates, the higher efficiency of our method compared to conventional Holder pasteurization was clearly demonstrated. The cumulative visualization of pressurization cycles vs. continuous pressurization presented in Figure 9 and Figure 10 demonstrates the reproducibility of both pressurization methods. As for the samples inoculated at UCT, the actual number of spores after milk inoculation was verified using several randomly selected samples with the assumption that all inoculated milk samples were equivalent if the same procedure was followed. Only samples with completed sets of measurements were used for statistical evaluation. For example, in experiment No. III, there was a loss of spores after milk inoculation, so the data from that experiment were excluded from statistical evaluation. Complete data are available at the Appendix A).

The authors present the results of their experimental study aimed exclusively at the evaluation of the high-pressure treatment on *B. cereus*, that can cause severe infection in premature babies [3,43,44,45,46,47,48]. This technology is an alternative to Holder pasteurization, which has been used for many years by Human Milk Banks as the standard method recommended by EMBA [49]. One of the reasons for its popularity is its effectiveness on the HIV, CMV, and HTLV viruses [50]. Moreover, its effectiveness on the Ebola and Zika viruses was recently demonstrated [51,52].

However, Holder pasteurization is ineffective against *B. cereus* spores [53]. Demazeau et al. has noted the effectiveness of the high pressure of 350 MPa on *B. cereus* spores [2]. The long-term experience of Human Milk Banks [49] as well as the results of our previous studies [5,6] showed that pasteurized milk safety can be achieved if routine post-pasteurization evaluation is performed and if samples (bottles) with positive microbiological findings are discarded. The total discard rate due to microbial positivity in our 5-year follow-up ranged between 8 and 10%, with spore-forming microbes accounting for almost 72% of all positive findings [5]. This was lower than that described by other authors [48,54]. The proportion of *B. cereus* contaminated milk in our Bank was also lower than that described by other authors [46,47,55]. The discard of the pasteurized milk sometimes led to temporarily lowered milk availability. In the case of the Milk Bank UH HK, financial losses can reach up to almost half a million CZK (~20,000 €) per year.

Despite the high discard rate, the amount of milk delivered annually from our bank is sufficient to cover the needs of Pediatric Intensive Care Units [5,6]. However, the discard rate has prevented the use of banked milk for a broader range of newborns, including healthy babies that cannot be breast-fed.

Our previous study showed that the quantity of *B. cereus* in discarded pasteurized milk was low, ranging from 1 to 100 CFU/mL [6]. The values below 10 CFU/mL, which is the limit of initial post-pasteurization *B. cereus* contamination used in Italy, Sweden, and UK [49], were found in 80% of cases [5,6]. Our initial *B. cereus* post-pasteurization limit is below 1 CFU/mL [6], and the same limit is also used in France, Australia, and the USA [49]. This same limit is also planned for use in clinical applications using the high-pressure treatment of human milk in the future.

If the quantitative data from our previous studies mentioned above are taken into account, high-pressure treatment would lead to a reduction in the number *B. cereus* spores below 1 CFU/mL limit and/or to negativity in the majority of post-high-pressure treatment bacteriological assessments. By accepting samples with a final “negative” result, we cannot know with certainty that the samples are free of *B. cereus* spores [49]. For this reason, the manipulation of high-pressure treated milk should be similar to the manipulation of pasteurized milk [6], preferably stored frozen and used within 1 h after thawing and warming to 37 °C.

High-pressure treatment has some disadvantages when compared to Holder pasteurization. While it has been shown to be effective on HIV and CMV viruses [10,56], its effect on other viruses such as Zika or Ebola has not yet been studied [17].

Table 6 lists the advantages and disadvantages of both methods, i.e., Holder pasteurization and high pressurization. The Table shows that the high-pressure method is more effective on spore-forming microbes, and its use could lead to financial savings. At the same time, it retains more bioactive substances [1]. A detailed review comparing both methods was published by Weselowska et al., 2019 [17], and the results of our study will be published later.

The use of Holder pasteurization is prescribed in the Czech Republic by a Decree of the Ministry of Health; however, this does not prevent the introduction of innovative methods if their efficiency is demonstrated relative to EU rules.

The processing of breast milk is currently regulated at the European Union level by European Parliament and Council Regulations No. 852/2004 and No. 178/2002, and by European Commission Regulation No. 2073/2005 [57,58,59], which mandate the use of procedures based on Hazard Analysis and Critical Control Points (HACCP). Microbiological criteria can be used in the validation and verification of procedures based on HACCP principles and other hygiene control measures. However, a legislative change is now being prepared at the European Union level, based on a Recommendation by the European Directorate for the Quality of Medicines & HealthCare, where breast milk should be classified as a substance of human origin (SOHO) [38]. From this point of view, high-pressure treatment could be introduced into common practice, provided that the method is directly validated for milk banking conditions. Therefore, the high-pressure method should be compared along with Holder pasteurization. We fully agree with the statement by Wesolowska that human participant studies are needed to assess promising new human milk processing techniques [17].

## 5. Conclusions

High pressurization methods are more effective against *B. cereus* spores compared to classic Holder pasteurization, and the application of pressure over several cycles is even more effective. On a set of 108 breast milk samples, we verified the effectiveness of two HPTs: (1) 350 MPa/5 min/38 °C in four cycles and (2) cumulative pressure of 350 MPa/20 min/38 °C. These methods reduced the number of spores by 3–4 orders of magnitude; HPT is more effective in cycles. Additionally, we verified that the method was reproducible. The implementation of this technique into routine practice could lead to an increased availability of breast milk for newborn babies, and at the same time, reduce the costs associated with discarded contaminated milk.

For common practice of the Human Milk Bank, we recommend using HPT of 350 MPa/5 min/38 °C in three cycles.

## Figures and Tables

**Figure 1 foods-12-04245-f001:**
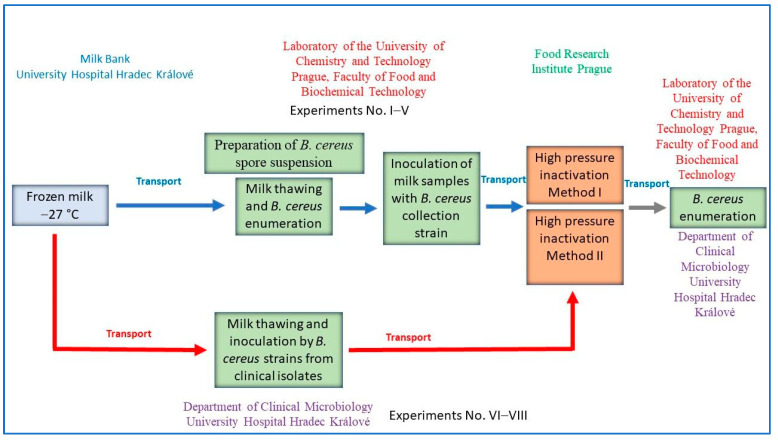
The algorithm of the experiment.

**Figure 2 foods-12-04245-f002:**
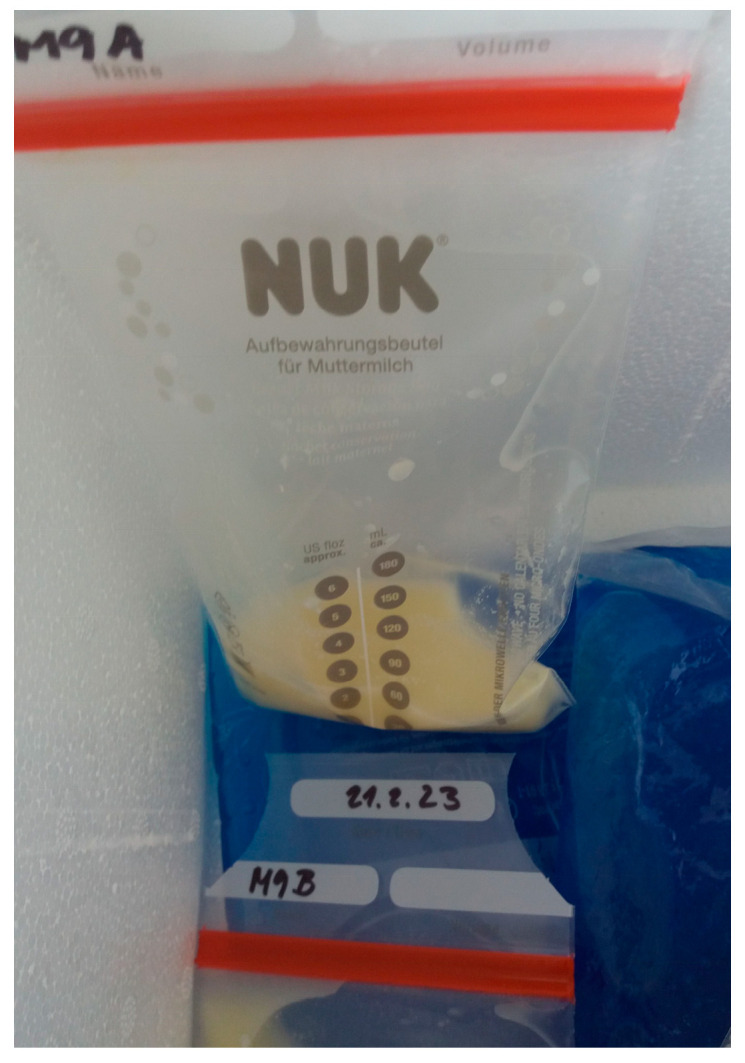
Plastic bags with milk samples intended for pressurization.

**Figure 3 foods-12-04245-f003:**
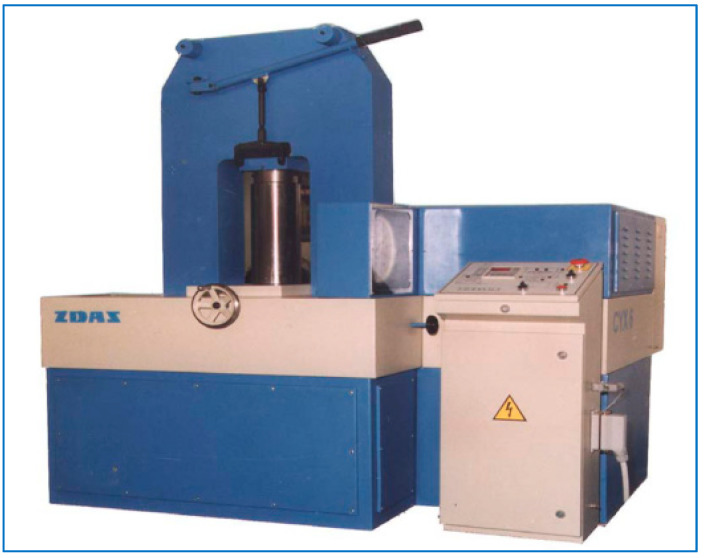
The high-pressure isostatic press CYX 6/103 (Žďas join-stock company, Žďár nad Sázavou, Czech Republic).

**Figure 4 foods-12-04245-f004:**
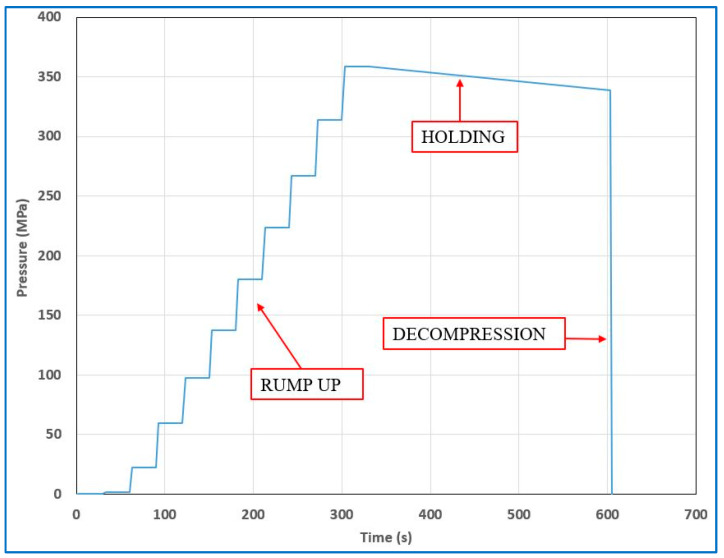
Pressurization cycles—pressure changes during one cycle.

**Figure 5 foods-12-04245-f005:**
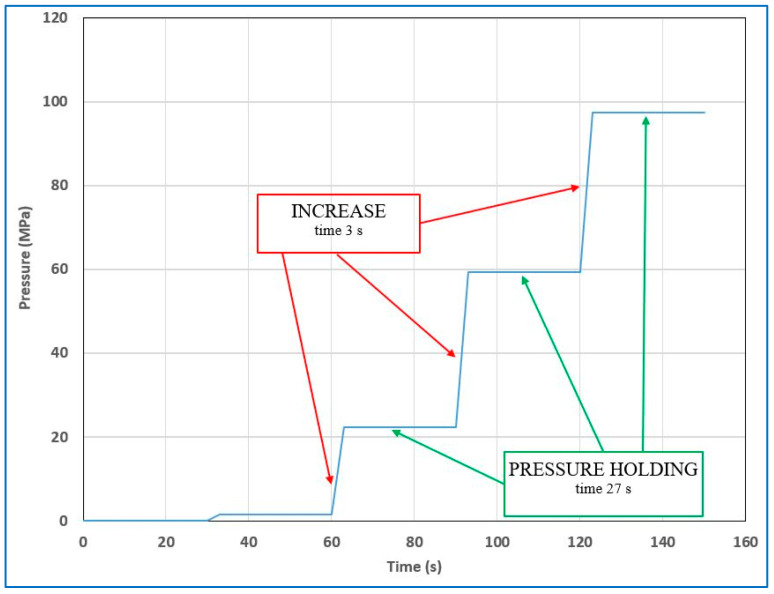
Pressurization cycles—detail changes.

**Figure 6 foods-12-04245-f006:**
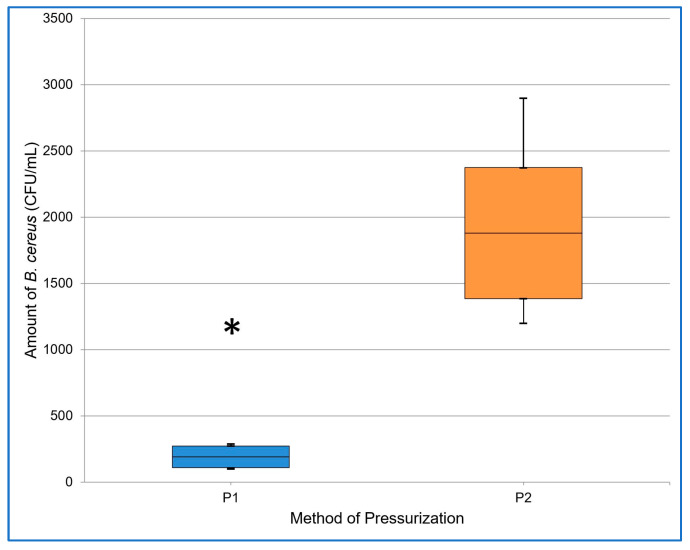
The resulting amount of *B. cereus* (CFU/mL) after applying pressure in cycles (P1) and continuous pressure (P2) in experiment I. The midline of the boxplot denotes the mean, the top line + SD, the bottom line—SD, and the whiskers denote the maximum and minimum values. The asterisk indicates the statistically significant results.

**Figure 7 foods-12-04245-f007:**
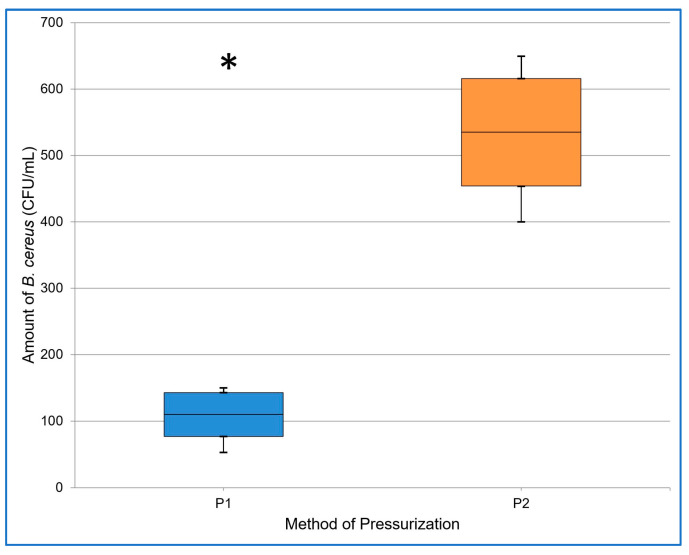
The resulting amount of *B. cereus* (CFU/mL) after applying pressure in cycles (P1) and continuous pressure (P2) in experiment II. The midline of the boxplot denotes the mean, the top line + SD, the bottom line—SD, and the whiskers denote the maximum and minimum values. The asterisk indicates the statistically significant results.

**Figure 8 foods-12-04245-f008:**
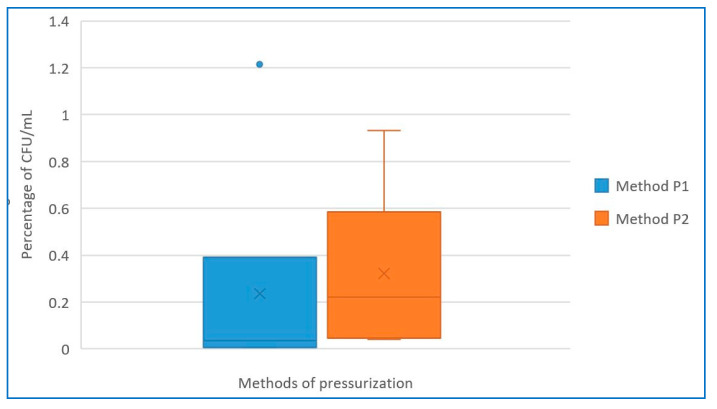
Residual percentage of microbial contamination (CFU/mL) after application of both pressurization methods. P1 denotes pressurization in cycles, P2 denotes continuous pressurization. The blue point in the graph represents an outlier.

**Figure 9 foods-12-04245-f009:**
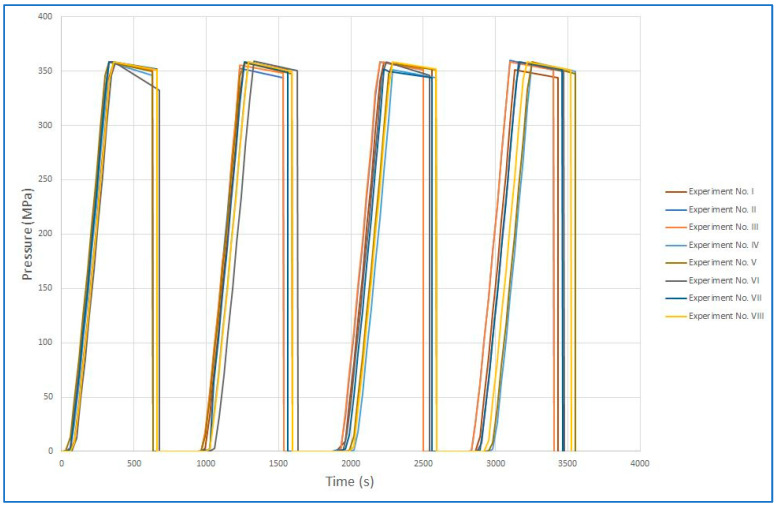
Cumulative visualization of pressurization in cycles.

**Figure 10 foods-12-04245-f010:**
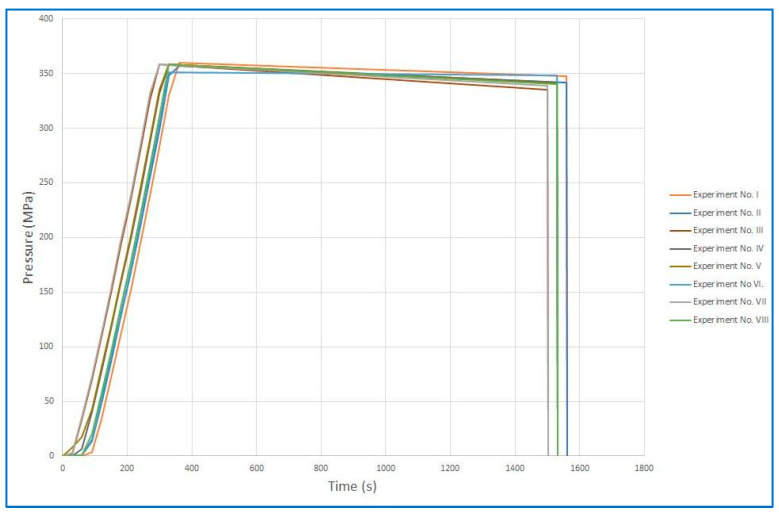
Cumulative visualization of continuous pressurization.

**Table 1 foods-12-04245-t001:** Chronological course of sample pressurization in four cycles with successive sampling in experiments No. IV and No. V.

	Cycles, Pressure 350 MPa, 38 °C			
Sample No.	Cycle 1	Cycle 2	Cycle 3	Cycle 4
1 and 2	5 min			
3 and 4	5 min	5 min		
5 and 6	5 min	5 min	5 min	
7 and 8	5 min	5 min	5 min	5 min

**Table 2 foods-12-04245-t002:** Results of the Two-Sample Equal-Variance *T*-Test from data for both methods of pressurization. P1 denotes pressurization in cycles, P2 denotes continuous pressurization, *n* denotes number of evaluated samples. The asterisks denote statistically significant differences.

Experiment No.	I		II	
Initial Count of *B. cereus* Spores CFU/mL	10^6^		10^5^	
Method	P1	P2	P1	P2
*n*	10	10	10	10
Minimum CFU/mL	100	1200	53	400
Maximum CFU/mL	290	2900	150	650
(Mean ± SD) CFU/mL	191 ± 82 *	1880 ± 494	110 ± 33 *	535 ± 81

**Table 3 foods-12-04245-t003:** Percentage of *B. cereus* CFU/mL after pressurization. P1 denotes pressurization in cycles, P2 denotes continuous pressurization, *n* denotes number of evaluated samples. The replicates are provided in the Appendix A.

Method	P1	P2
*n*	6	6
Minimum CFU/mL	0.01	0.04
Maximum CFU/mL	1.21	0.93
(Mean ± SD) CFU/mL	0.24 ± 0.48	0.32 ± 0.34

**Table 4 foods-12-04245-t004:** Descriptive statistics for samples inoculated at UH HK after both methods of pressurization (Experiments No. VI–VIII). P1 denotes pressurization in cycles, P2 denotes continuous pressurization, *n* denotes number of evaluated samples. The asterisks denote statistically significant differences.

Method	P1	P2
*n*	28	28
Median (Q1; Q3)	0 (0;0)	0 (0;0)
(Mean ± SD) CFU/mL	0.04 ± 0.19 *	0.36 ± 0.95

**Table 5 foods-12-04245-t005:** The final amount of *B. cereus* spores after high-pressure treatment in Experiment No. IV and V. P1 denotes cycled pressurization and P2 denotes continuous pressurization.

		The Amount of *B. cereus* Spores in CFU/mL	
Experiment No.	Sample No.	P1	P2
IV	1	1.7 × 10^2^	
	2	1.4 × 10^2^	
	3	25	
	4	10	
	5	10	
	6	5	
	7	5	
	8	<5	
	9		<5
	10		20
	BLANK 1	2.9 × 10^6^	2.4 × 10^6^
	BLANK 2	<5	<5
V	1	4.3 × 10^3^	
	2	5.3 × 10^3^	
	3	1.6 × 10^3^	
	4	1.1 × 10^3^	
	5	5.1 × 10^2^	
	6	4.5 × 10^2^	
	7	2.7 × 10^2^	
	8	1.8 × 10^2^	
	9		1.3 × 10^3^
	10		1.1 × 10^3^
	BLANK 1	5.4 × 10^2^	6.7 × 10^6^
	BLANK 2	1.2 × 10^2^	2.30 × 10^6^

**Table 6 foods-12-04245-t006:** A comparison of Holder pasteurization and high pressurization inactivation [5,27].

Parameter	Holder Pasteurization	High-Pressure Inactivation
Duration	30 min	15–20 min
Temperature	62 °C	38 °C
Spore inactivation	No	Yes
Gentleness to bioactive substances	No	Yes
Device operation	less demanding	more demanding

## Data Availability

The data presented in this study are available on request from the corresponding author.

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
