# Peer review of "High-Pressure Inactivation of Bacillus cereus in Human Breast Milk"

_foods, 2023, doi:10.3390/foods12234245_

Round 1

Reviewer 1 Report

Comments and Suggestions for Authors

Please check the corrections in the paper.

Author Response

Dear Reviewer,

thank You very much for Your valuable comments that can help me to improve the manuscript. Please see my corrections in the uploaded file.

Your sincerely Miroslava Jandová

Reviewer 2 Report

Comments and Suggestions for Authors

High pressure inactivation of Bacillus cereus in human breast milk

L 38: Mention this method preserves the nutritional quality of milk as well.

Please elaborate on whether this technique increases the shelf life or shelf stability of milk. How many folds? The shelf life of this milk is ____ days or ___ weeks etc.

L 45: Add a citation, please.

46. Why capital?

L 49: Mention the holder pasteurization conditions.

L 60. Please also add few reference for high pressure processing, check these articles:

Innovations in High-pressure Technologies for the Development of Clean Label Dairy Products: A Review

L 90: The results shoved or showed? Check this.

L 114: Mention the sample size.

L 127: Mention the centrifuge machine used.

L 129, 151: Kindly add a reference here.

L 206, 221: Kindly add a citation for the protocol followed in the study.

L 252, 261: Re-check the standard deviation in Tables 2 and 3, please.

L 332: Mention the pressure conditions. Please.

L 342: Add a reference, please.

L 393: The shelf life increase should be mentioned here.

In references, the page numbers range is missing.

Please avoid short sentences, merge with previous or next paragraphs.

Comments on the Quality of English Language

It is fine.

Author Response

Dear Reviewer,

Thank You very much for Your valuable comments that can help me to improve the manuscript. Please, see my corrections in the uploaded file.

Your sincerely Miroslava Jandová

Reviewer 3 Report

Comments and Suggestions for Authors

The manuscript presents a study to investigate the high-pressure methods for the inactivation of Bacillus cereus in human breast milk. High pressure in cycles method was compared with the cumulative high pressure method and the first one was demonstrated to be more effective. However, there are many issues that need to be addressed.

All Bacteria name needs to be italicized. The abbreviation name of bacteria should be provided the first time using the full name.

Line 52, what does it mean by "biological quality"? Needs to be more specific.

Line 59, " HPT should preserve the high quality of milk". Are there any references to support the statement here? Why can HPT preserve the quality?

Line 86, what is the unit for 10^6?

Line 65-75, many previous works using high pressure have been described. What is the research gap of current HPT? Needs more clarification.

The novelty of the work needs to clarified. Since so many studies have been done on HPT, what is the novelty of this study?

Line 111-113, the description should be in the introduction section, not in the materials and methods section.

Figure 1 is not necessary for the manuscript's main text. Can be moved to supporting information.

Section 2.1.2, Is this part B. cereus vegetative cells or spores determination? Needs to be more specific in the sub-title.

What is the difference of section 2.1.3 and 2.2.1? The sub-titles are the same.

Suggest moving Fig2 and 3 to supporting information.

What is the meaning of n in Table 2?

Table 3, The results of each replicate should be in the supporting info.

Figure 6, the meaning of method P1 and P2 should be described in the caption.

The figures and tables should be revised in the manuscript. More figures about the results should be included for data visualization. The tables with more detailed data should be in the supporting info. 

Comments on the Quality of English Language

Minor changes regarding the quality of English language need to be made.

Author Response

Dear Reviewer,

thank You very much for Your valuable comments that can help me to improve the manuscript. Please, see my corrections in the uploaded file.

Your sincerely Miroslava Jandová
